# Policy Responses to the COVID-19 Pandemic in Vietnam

**DOI:** 10.3390/ijerph18020559

**Published:** 2021-01-11

**Authors:** Tuyet-Anh T. Le, Kelly Vodden, Jianghua Wu, Ghada Atiwesh

**Affiliations:** 1School of Science and the Environment, Grenfell Campus, Memorial University of Newfoundland, Corner Brook, NL A2H 5G4, Canada; jwu@grenfell.mun.ca; 2Forestry Economics Research Centre, Vietnamese Academy of Forest Sciences, 46 Duc Thang Ward, Northern Tu Liem District, Hanoi 11910, Vietnam; 3Environmental Policy Institute, Grenfell Campus, Memorial University of Newfoundland, Corner Brook, NL A2H 5G4, Canada; 4Department of Environmental Science, St. John’s Campus, Memorial University of Newfoundland, NL A1C 5S7, Canada; ga1871@mun.ca

**Keywords:** coronavirus, COVID-19 in Vietnam, pandemic in Vietnam, policy documents, policy tools, policy responses, policy measures, Vietnam

## Abstract

The COVID-19 pandemic has become one of the most serious health crises in human history, spreading rapidly across the globe from January 2020 to the present. With prompt and drastic measures, Vietnam is one of the few countries that has largely succeeded in controlling the outbreak. This result is derived from a harmonious combination of many factors, with the policy system playing a key role. This study assessed the policy responses to the COVID-19 pandemic in Vietnam from the early days of the outbreak in January 2020 to 24 July 2020 (with a total of 413 cases confirmed and 99 days of no new cases infected from the local community) by synthesizing and evaluating 959 relevant policy documents in different classifications. The findings show that the Vietnamese policy system responded promptly, proactively, and effectively at multiple authority levels (33 different agencies from the national to provincial governments), using a range of policy tools and measures. Parallel to the daily occurrence of 2.24 new cases, 5.13 new policy documents were issued on average per day over the study period. The pandemic policy response over the first six months in Vietnam were divided into four periods, I (23 January–5 March), II (6–19 March), III (20 March–21 April), and IV (22 April–24 July). This paper synthesizes eight solution groups for these four anti-pandemic phases, including outbreak announcements and steering documents, medical measures, blockade of the schools, emergency responses, border and entry control measures, social isolation and nationwide social isolation measures, financial supports, and other measures. By emphasizing diversification of the policy responses, from the agencies to the tools and measures, the case study reviews and shares lessons from the successful COVID-19 prevention and control in Vietnam that could be useful for other nations.

## 1. Introduction

The first confirmed case (CC) of the COVID-19 disease outside of China appeared in Thailand on 13 January 2020 [1,2]. Just 18 days later, on 30 January, in the first report on the epidemic situation, the World Health Organization (WHO) claimed that there were a total of 7818 global confirmed cases (CCs) with the majority coming from China, and only 18 cases from 18 other countries [1]. Globally, as of 24 July 2020, the COVID-19 coronavirus had spread in 213 countries and territories, with 15,296,926 CCs and a death toll of 628,903 individuals [3]. Compared to the number of CCs on 30 January, the 24 July number had increased 1956.63 times over nearly six months. The date of 24 July also marked the day with the highest number of new cases (NCs) in the past six months (284,196 NCs) [3,4,5]. COVID-19 deaths also reached a peak on 24 July of 9753 deaths since a record high of 9797 deaths on 30 April [3,4,5]. In contrast, no one had died due to COVID-19 in Vietnam, and the number of recovered cases (RCs) had reached 365 patients (88.38%) of the total of 413 CCs, as of 24 July. Only 48 cases were being treated (11.62%) [6,7]. As of this date, 99 days in Vietnam had been spent without any NCs infected from the local community [8,9]. In this study, we offer an overview of the pandemic prevention and control in Vietnam, which was successful and effective, up to 24 July. The overview was divided into four periods: I (23 January–5 March), II (6–19 March), III (20 March–21 April), and IV (22 April–24 July) [10]. Within the scope of this study, the CCs and policy responses and actions put in place to minimize infection and spread over the initial six-month term from 23 January to 24 July were gathered, analyzed, and assessed.

With CC number (#) 416 on 25 July [11,12,13,14], the 100-day series of COVID-19 non-infectiousness in Vietnam was terminated [11]. This marked the beginning of period V, with complicated and unpredictable developments in terms of both the pandemic [15] and social safety due to a series of Chinese people entering Vietnam illegally through border roads and waterways [16]. As of 15 December, the whole country had recorded 35 deaths and 1405 CCs [6]. From 25 July to 15 December (144 days) an additional 992 NCs were reported—equivalent to 6.89 NCs/day on average, in contrast to the 2.24 CCs/day in previous periods (184 days with 413 CCs). In the first days of period V, there were up to dozens of NCs daily for many consecutive days. Period V also saw 35 deaths, following a period of no deaths because of the coronavirus in all four previous stages [17]. The main cause was identified as a mutation of the virus, leading to infection in the family and high cross-infection [18]. This was the sixth identified strain of COVID-19 in Vietnam, with the characteristic of spreading faster than previous strains recorded [16]. As of 15 December, Vietnam had witnessed 79 days with no more deaths, remaining at a confirmed number of 35 since 3 September [6]. 

Therefore, a question arises: Why in so many nations and territories across the world, including the most wealthy countries such as the USA, Brazil, France, the UK, and Germany, did the pandemic spread and cause hundreds, even thousands of deaths daily [19,20], whereas in Vietnam—a small, densely populated neighbor of China with only 331,698 km^2^ land area [21] and over 97.43 million people [22] that is a developing country with limited resources—was able to effectively control the COVID-19 pandemic [19,23,24]? Many wealthier countries have praised Vietnam’s response to COVID-19 [20]. George Black—a writer in New York City (USA) has considered that “Vietnam may have the most effective response to the COVID-19 through mass mobilization of the health care system, public employees, and the security forces, combined with an energetic and creative public education campaign” [25] whereas Fages showed that four factors, including quick strategic testing, aggressive contact tracing, an effective public communications campaign, and swift development of testing kits, have led to other countries praising Vietnam [20]. In some studies, Vietnamese scholars pointed to the prompt and effective policy responses as one of the most important factors contributing to COVID-19-related successes in Vietnam. The study by La et al., for example, demonstrated that timely policy responses to the outbreak from the government and the media, integrated with updated research on COVID-19, together provided reliable sources of information [26]. Another survey conducted by Lê and Nguyễn also demonstrated that many countries and international organizations praised Vietnam for the resolute and prompt solutions of governments to prevent and control the pandemic, including the effective measures in quarantining and classifying people from pandemic zones [27]. This study offers an overview of the policy responses to the COVID-19 pandemic in Vietnam from the preparation phase on 16 January (before the first 2 CCs on 23 January) to 24 July by synthesizing and evaluating the relevant policy system in different classifications, including by agencies, time periods, types of policy communication, and category of policy responses.

## 2. Materials and Methods

Data on CCs were compiled from https://ncov.moh.gov.vn/web/guest/trang-chu—the website of the Ministry of Health (MOH) in Vietnam on the COVID-19 pandemic status updates, as well as the related study by La et al. [26]. We further report findings from collecting, classifying, and synthesizing 959 COVID-19 policy documents (PDs) from different government levels from central (national) to provincial. These PDs were gathered through two main channels: (1) https://thuvienphapluat.vn/ (legal library)—the most powerful website on Vietnamese legal documents--and (2) https://ncov.moh.gov.vn/web/guest/chinh-sach-phong-chong-dich—a portal opened on the MOH’s website on the COVID-19 pandemic prevention and control policies.

Other relevant documents were also obtained from database analysis of recent policies, official press, articles, reports, briefs, and presentations from reliable data sources in Vietnam and globally. A total of 236 documents were obtained during the period of analysis.

Within our research, PDs under the provincial level (districts and communes) were not collected for analysis. Although at the 13th National Assembly session, the district and commune authorities officially obtained the right to issue legal documents after 22 June 2015 [28], so far, there is no official channel for storing COVID-19 PDs promulgated by district and commune levels.

After these documents were identified and collected, the analysis steps included (i) mapping a chronology of all CCs from the first day of outbreak in Vietnam (23 January) to the last day of the term (24 July) in each of four periods of the pandemic development in Vietnam, as declared by the Prime Minister (PM): I (23 January–5 March), II (6–19 March); III (20 March–21 April), and IV (22 April–24 July) (findings are in Section 3.1); (ii) classifying and synthesizing the PDs released by 33 different agencies and by periods (findings in Section 3.2) and PD types (resulting in the identification of nine key categories of PDs) (findings in Section 3.3); and lastly analyzing the evolution of policy according to the four periods of the pandemic development (findings in Section 3.4).

## 3. Results

### 3.1. Chronology of New and Confirmed Cases

Figure 1 and Appendix A show a timeline of the spread of COVID-19 in Vietnam, tracing from the first two CCs on 23 January to CC #413 on 24 July 2020. The data sources of this study from CC #1 (on 23 January) to CC #207 (on 31 March) have been drawn and then checked from La et al. [26] and from CC #208 (on 1 April) to CC #413 (on 24 July) from the MOH’s website [6]. Corrections were made to some inaccurate information points about 4 CCs in La et al.’s study [26], including the appearance of two CCs on 3 February 2020, #8 [29] and #9 [30], not only one CC, #8, as suggested by La et al. [26] and corrections to the nationalities of CCs #151, #152, #186, and #207.

The classifications of COVID-19 patients and direct contacts are defined from F0 to Fi (i = 1, 2, 3, or more). To specify, F0 people are confirmed as infected, F1 people are suspected as infected or contacted F0, and F2 people contacted F1 and similarly the next generations F3, F4, etc. [31,32]. The map of 413 CCs (Figure 1) illustrates that with timely and drastic government policy responses, the rate of infection after F1 strongly declined. The whole country has 24 F0 CCs (5.81%), which resulted in 48 F1 CCs (11.62%). This led to 12 F2 CCs (2.91%) out of a total of 413 CCs. In each of the F3 and F4 generations only one CC appeared, which originated from CC #133—one of the super infectious cases and the fastest spreading case in period III (20 March–21 April)—the time with the highest number of cases of infection in the country [10].

24 July 2020 was the 184th day of fighting against the COVID-19 pandemic in Vietnam since the first two cases on 23 January. During these 184 days, there was a total of 413 CCs nationwide. Over more than six months of combat, an average of 2.24 NCs per day were detected. There were 365 RCs [7] out of 413 CCs [33] (accounting for 88.38%), of which the first 16 CCs were completely recovered, 349 CCs from 6 March to 20 July were recovered, and 52 CCs were being actively treated (12.47%) [6]. According to statistics of the MOH [6], of the 413 CCs, 70 (16.95%) were foreign nationalities (of which 49 had recovered with treatment by Vietnamese doctors) and 343 were Vietnamese CCs (including 237 people having returned from abroad).

The first period in the fight against COVID-19 in Vietnam consisted of 16 patients confirmed in 43 days (23 January–5 March, 2020), corresponding to 0.37 CC/day, who all had a history of movement from China—the country where the outbreak first appeared [34], with two Chinese, one American, and 13 Vietnamese patients [6]. The second period lasted two weeks (6–19 March) and confirmed a total of 69 cases, corresponding to 4.93 CCs/day on average, 13.25 times higher in comparison to the first period. Within period I there appeared one special CC (#5) on 30 January, who was one of three first Vietnamese CCs of an eight-member delegation sent by Nihon Plast Co., Ltd. (Vinh Phuc province) to train in Wuhan city for 2.5 months. CC #5 then infected six other CCs in Vinh Phuc [35,36]. Meanwhile on 6 March, Hanoi announced the first CC (#17) returning from England [10,37], who infected four CCs (#19, #20, #32 and #54) (see Figure 1). With CC #17, Vietnam ended a consecutive string of 22 days without any NCs and officially entered the second phase with CCs coming from abroad [37,38]. The most super contagious case (SCC) in this period was CC #34 who returned from Washington (USA) to Ho Chi Minh city (HCMC) on 10 March, and then infected 11 other cases [37], including eight F1 CCs and three F2 CCs (see Figure 1). 

Compared to periods I and II, period III (20 March–21 April) occurred over the highest number of 33 consecutive days with recorded NCs due to positive test results with the new strains of COVID-19 from both Vietnamese and foreign passengers from the flights from period II, still having flights to welcome repatriated Vietnamese, and especially outbreaks of infectious chains in the community. This phase recorded a total number of 183 CCs (5.55 CCs per day on average), of which there were up to 54 CCs related to infectious chains in the community (accounted for 29.51%). Notably, this period had three SCCs, #124 and #133 on 24 March and #243 on 6 April (see Figure 1). On 24 March, with the detection of CC #124 (infected from Buddha bar—the location of the highest infection in HCMC with a total of eight CCs as of 25 March and closed since 21 March [39,40,41,42], it was discovered to be the source of infection for 11 F1 people and 19 F2 people, which resulted in three F1 CCs and one F2 CC [43]. The date of 24 March was also the day of detection of CC #133, which was considered the most SCC by creating four COVID-19 CC generations with three F1 CCs out of a total of 43 F1 people who were in contact with CC #133, two F2 CCs out of a total of 699 F2 people who were in contact with some of the 43 F1 people and one F3 CC out of a total of more than 2100 F3 people and one F4 CC [40,41]. The date of 6 April recorded another SCC—CC #243, who had been treated at Bach Mai hospital [44,45] and then infected four F1 CCs and five F2 CCs (see Figure 1 and Appendix A).

Period IV existed for 94 days (22 April–24 July) and confirmed a total of 145 cases (1.54 CCs/day) and 280 PDs (see Figure 1 and Appendix A). The most significant and widely noted aspects of stage IV were, first, that the pandemic was well controlled [46], with all CCs originating from flights welcoming returning overseas Vietnamese. As of 24 July, Vietnam had experienced 99 days without new infections from the community [8]. Second, there was an acceptance of the situation of living with the pandemic, to move to a longer, more radical, anti-outbreak stage along with socio- economic development [47].

In brief, over the first six months of combatting COVID-19, Vietnam detected an average of 2.24 CCs and 1.98 RCs per day, which means that each day added 0.26 more NCs to treat. This indicator can be compared to other countries considered to have strong pandemic management in the first stages, such Taiwan, Hong Kong, Singapore, and South Korea [48,49,50]. Databases of international statistics organizations [51,52] demonstrate that these figures in Taiwan and Hong Kong were 0.1 and 5.22, respectively [48,49,50,51,52,53]. As of 24 July, Taiwan had spent 185 days in the outbreak and detected 458 CCs [51,52], corresponding to 19 CCs per 1 million population [51] and 2.48 CCs per day. These figures in Vietnam were only 4 CCs/1 million population [51] and 2.24 CCs/day. These typical indicators illustrate that Vietnam attained significant achievements in the global fight against the COVID-19 pandemic. Despite an increase in the second COVID-19 pandemic wave from the last days of July to September, these successes in the first wave of the COVID-19 pandemic remain notable and offer lessons for fighting COVID-19 and future pandemics.

### 3.2. Urgent and Timely Policy Responses by All Relevant Authorities

Vietnam carefully prepared to respond to the pandemic before the appearance of the first two positive cases from Wuhan, China, on 23 January, 2020 [26,45], which were confirmed at Cho Ray hospital in HCMC, just two days before the Lunar New Year holidays [34]. News of a “strange pneumonia” in China had circulated in Vietnamese media as early as the beginning of January 2020, before the first CCs [26]. The term “strange pneumonia” first arose on 9 January on baochinhphu.vn—the Vietnam Government Portal [54], and suckhoedoisong.vn—the official news channel of the Ministry of Health (MOH) [55], followed by a reference again in Vietnam News—the national English language daily (vietnamnews.vn) on 17 January [56].

Before 23 January, there were seven COVID-19-related official legal documents. The terms “pneumonia” and “novel coronavirus” were first announced widely on 31 December, 2019 [1]. In other words, although the information about the COVID-19 epidemic in December 2019 was not official, the MOH in Vietnam issued decision No. 5894, dated on 19 December 2019 about “Guidelines for the establishment and operation of Quick Response Teams (QRTs).” This document was designed to guide the establishment and operation of the QRTs from the national to the district level to respond promptly and effectively to infectious epidemics and public health events in the context of an increasing number of new and reappearing epidemics that have serious impacts on people’s health as well as global health security [57]. The first legal document related to the term “coronavirus” was officially released by the Government Office (GO) with dispatch No. 441 dated 16 January, 2020, about the pneumonia epidemic in China [58,59,60,61,62]. On the same day, the MOH expeditiously issued decision No. 125 on guidelines for the diagnosis and treatment of acute pneumonia caused by a new strain of coronavirus (nCoV) [63,64,65]. The next day, 17 January, there were two documents issued by the MOH, decision No. 137 on promulgating the plan for the prevention and control of the infectious epidemic in 2020 [66], and emergency dispatch No. 62 on early detection and good preparation for the prevention and control of the epidemic [65,66,67]. 

Immediately, the People’s Committee (PC) of Lang Son province also directed the response to the pneumonia epidemic from China [58]. Lang Son—one of seven provinces of Vietnam bordering China—is a northern province with the largest number of border gates to China in nine locations: Huu Nghi, Dong Dang, Chi Ma, Binh Nghi, Coc Nam, Po Nhung, Co Sau, Ban Chat, and Na Hinh [68]. From 20 to 22 January, each day witnessed one legal document released by this ministry: Decision No. 156 provided a plan to respond to the epidemic [69], decision No. 181 provided temporary guidelines for surveillance and prevention and control of the epidemic for hospitals and clinics nationwide [45,70,71], and directive No. 03 (dated 22 January) focused on strengthening the prevention and control of the epidemic [72]. These legal documents were the first ministerial documents and dealt directly with the core professional problems of providing diagnosis and treatment guidelines, plans for acute pneumonia that were published one week earlier than the first CCs in Vietnam. This points out the proactive nature and high level of preparedness to cope with the pandemic that existed in the country.

Figure 2 and Appendix A provide an overview of 413 CCs and 959 policy documents (PDs) in Vietnam from 13 January to 24 July. New PDs were issued almost every day. On average, each day 2.24 NCs appeared and 5.13 new PDs were issued as of 24 July 2020. Although only the first 16 cases coming from China [10,37] were confirmed in the 43 days of period I (23 January–5 March), the emergency PDs were issued promptly with 190 documents (4.42 PDs/day on average). In some early days there were up to 15–17 PDs/day. Meanwhile the 14 days of the second period (6–19 March) witnessed 69 CCs and—with CCs from abroad [10,37]—125 relevant PDs, corresponding to 4.93 CCs/day and 8.93 PDs/day on average, 13.25 times and over two times higher than those in the first period, respectively. During all 33 consecutive days of period III (20 March–21 April) 183 CCs (5.55 CCs/day) and 125 PDs (10.82 PDs/day) were recorded as the infection escalated in the country [10,37]. New PDs at all levels, especially at the provincial levels, were issued every day to strengthen the pandemic management in the new context of complicated developments. With the entire nation’s effort, the serious situation of period III led to period IV (22 April–24 July, 94 days), with policy concentrated on both fighting against the pandemic and developing strategies for dealing with relevant socio-economic consequences [10,37]. This period confirmed a total of 145 cases and 280 PDs, corresponding to 1.54 CCs/day—the second lowest figure in all periods, and 2.98 PDs/day—the lowest number in the whole time studied (23 January–24 July) (see Figure 2 and Appendix A).

#### 3.2.1. Policies by Levels and Agencies

To examine the levels and agencies in which policy actions were taken, four relevant policy levels were identified from the highest level (central/national government) to the lowest (provincial) (see Figure 3 and Appendix A). The classification of four PD levels of management is based on the organizational diagram of the State of the Socialist Republic of Vietnam in 2016 [73]. All PDs from this study were categorized by the issuing authority at each level.

Figure 3 and Appendix A illustrate the number of Vietnamese COVID-19 PDs of each authority, as well as the qualitative relationships between PDs classified by authority levels and the number of CCs of each province, from the earliest days of the outbreak to 24 July. The information of central levels (CLs) is represented by the inner red circle, central focal agencies (CFLs) in purple, ministerial agencies in green, and provincial people’s committees (PPCs) in blue. Specifically, the central levels (CLs) include four units: the National Assembly Standing Committee (NASC), Government of the Socialist Republic of Vietnam (GOV), Prime Minister (PM), and Secretariat of the Communist Party of Vietnam (SCPV), and the central focal agencies (CFLs) cover four units: the Central Propaganda and Training Commission (CPTC), National Steering Committee for COVID-19 Prevention and Control (NSCPC), the Supreme People’s Court (SPC), and the Supreme People’s Procuracy (SPP). The ministerial agencies (MAs) include 18 ministries: (1) Ministry of Agriculture and Rural Development (MARD), (2) Ministry of Information and Communications (MIC), (3) Ministry of National Defense (MND), (4) Ministry of Construction (MOC), (5) Ministry of Culture, Sports and Tourism (MOCST), (6) Ministry of Education and Training (MOET), (7) Ministry of Finance (MOF), (8) Ministry of Foreign Affairs (MOFA), (9) Ministry of Health (MOH), (10) Ministry of Home Affairs (MOHA), (11) Ministry of Industry and Trade (MOIT), (12) Ministry of Justice (MOJ), (13) Ministry of Labor, Invalids and Social Affairs (MOLISA), (14) Ministry of Natural Resources and Environment (MONRE), (15) Ministry of Science and Technology (MOST), (16) Ministry of Transport (MOT), (17) Ministry of Planning and Investment (MPI), and (18) Ministry of Public Security (MPS). The ministerial-level agencies (MLAs) cover six units: (1) Committee for Ethnic Minority Affairs (CEMA), (2) the Government Office (GO), (3) the State Audit of Vietnam (SAV), (4) the State Bank of Vietnam (SBV), (5) Vietnam General Confederation of Labor (VGCL), and (6) Vietnam Social Security (VSS). The PPCs include all 63 provinces. Vietnam covers 63 provincial units with 61 provinces and two special urban units under central authority, including HCMC and Hanoi capital—equivalent to the provincial level [74,75].

One of the most distinctive features of the COVID-19 policy responses in Vietnam was the mobilization of the whole political system, accepting economic losses in exchange for the safety of people’s health and lives, minimizing deaths from the pandemic [76]. As of 24 July, all 18 ministries had issued 406/959 documents (42.34%) and all 63 provincial units throughout the country had announced and implemented their own PDs as well (413/959 documents, accounting for 43.07%). By 24 July 2020, Vietnam saw CCs in 36 out of 63 provincial units. In Hanoi and HCMC, where the two largest international airports, Noi Bai and Tan Son Nhat, are located, these provincial units had peak numbers of CCs (121 and 62, respectively) as well as the most significant quantities of legal documents (45 and 39, respectively) (see Figure 3 and Appendix A). 

#### 3.2.2. Timing of Policy Responses

The numbers of PDs were also identified by the five types of agencies (CLs, CFLs, MAs, MLAs, and PPCs) and by period. The multi-level policy approach came from 33 different agencies (from the national to provincial levels), classified in five groups (including 4 CL units, 4 CFL units, 18 MA units, 6 MLA units, and 63 PPC units), in all periods from the beginning days of the outbreak (see Figure 4 and Appendix A).

Figure 5 and Appendix A illustrate the result of a review of the number of PDs classified by 33 different agencies from the early days of the outbreak to 24 July 2020. The whole process saw 959 COVID-19 PDs released, with 7 in the preparation period, 190 in the first period, 125 in the second, and 357 and 280 in the third and the fourth, respectively, from the central government to the ministry levels and a group of PPCs (including all 63 provincial units nationwide). The PPCs released the greatest number, 314/959 PDs (32.74%), followed by the MOH with 138/959 documents (14.39%), and the MOET and the GO with 8.86% and 7.51%, respectively (see Figure 5 and Appendix A). One of the findings of La et al.’s study, based on an analysis of COVID-19 news reports in Vietnam from the early days to April 4th, 2020, was that 173 official instructions, guidelines, plans, dispatches, policies, and direct actions were issued by the central government and the relevant ministries [26]. However, our study demonstrates that there were 424 COVID-19 official legal documents, from the central government to the MLs and MLAs, as of April 4th, (see Figure 2), a number 2.45 times higher. With the highest number of PDs in period I from the CLs (15), this analysis demonstrates that the political system of Vietnam put in place strong prevention and control measures from the early days. This extended from the central to local levels throughout the country.

### 3.3. Diversification of Policy Types

Not only did many government agencies promulgate PDs quickly, but each authority also issued different types of PDs. This study identified nine types of COVID-19 legal documents used in Vietnam as of 24 July 2020 (see Figure 6 and Appendix A), each of which are defined in Appendix B (see Box 0). According to this review, COVID-19 PDs have not been categorized by previous authors.

Of the 959 PDs created during the COVID-19 pandemic in Vietnam up to 24 July, dispatches were the most used policy communication type, with 577 dispatches (60.17% of the total # of documents), followed by 115 decisions (11.99%), 75 plans (7.82%), and 74 directives (7.72%). Other documents with lower rates included 51 announcements (5.32%), 32 telegrams (3.34%), 27 resolutions (2.82%), 5 circulars (0.52%), and 3 guidelines (0.31%) (see Figure 6 and Appendix A).

Analyzing the PD types used by various agencies, dispatches were again the most widely used; 24/33 agencies chose to make their policies known in this way, followed by decisions (14/33), telegrams (12/33), directives (10/33), and announcements (7/33). The lowest ranks belonged to circulars and guidelines, with each used only by two agencies (see Figure 7 and Appendix A).

Among the 33 agencies, PPCs (from the 63 provincial units) not only utilized the most diverse range of policy document types (8/9), but in each type (except circulars and announcements) they also accounted for the largest proportion of the communications generated in comparison with the other agencies, from 28.13% of telegrams to 96% of plans, as illustrated in Figure 7. The next most diverse set of PDs were released by the MOF, using 6/9 types. The SBV and the MOH each used 5/9 types. Most other authorities issued 2–4 different types, such as the MOET (4/9), the PM, the GO, the NSCPC, the MOCST, and the MOLISA (each using 3/9), and the MOC (2/9) (see Figure 7 and Appendix A).

### 3.4. Diversification of Policy Measures

The policy responses to the COVID-19 emergency have varied between countries and there have been different methodologies used to assess the nature and role of COVID-19 policies across the world. The Blavatnik School of Government at the University of Oxford, for example, has been using a “stringency index” based on seven policy measures to compare countries’ differing COVID-19 policies. They include school closings, workplace closings, cancellation of public events, shutting down of public transport, public information campaigns, restrictions on internal movement, and controls on international travel [77]. The School’s research focused on 44 countries, but without Vietnam. Another methodology to assess country responses was built by the team at Eurasia Group across three areas: healthcare management, political response, and financial policy response. This study included 11 key countries, again excluding Vietnam: Taiwan, Singapore, South Korea, New Zealand, Australia, Canada, Germany, Iceland, United Arab Emirates, Greece, and Argentina [48].

The COVID-19 policy measures in Vietnam have also attracted the attention of many foreign scholars. In a study of the Exemplars in Global Health platform, Vietnam’s emerging COVID-19 success story as of 1 July, 2020, was identified as coming from six policy groups: rapid developed diagnostic test kits; contact tracing; infection prevention and control in healthcare settings; targeted lockdowns; mass gathering, travel, and mobility restrictions; and clear, consistent, creative public health messaging [78]. Some of these findings are similar to those of other scholars [20,25,79,80]. Some Vietnamese scholar groups have classified the COVID-19 policy responses in Vietnam according to specific content, such as fake news prevention, assessment of the prevention or of the threat, education, emergency response, market control, national funding, preventive action, reward, social distancing announcement, citizen support, and travel restrictions [26]. The Vietnamese State and the national government have taken 13 key measures to prevent and control the COVID-19 pandemic, including (1) bringing Vietnamese citizens back from the pandemic areas, (2) constructing field hospitals for isolation, (3) utilizing universities’ facilities as concentrated isolation zones, (4) keeping all students from schools to limit community contact, (5) issuing bans on crowded meetings and suspending business and entertainment activities for a limited period during the peak period of the pandemic, (6) regularly sending messages via social networks and individuals’ mobile phones from the government and the MOH, (7) creating applications on mobile devices to provide information relating to the COVID-19 pandemic, (8) conducting all-people medical declarations related to COVID-19, (9) strict control of entry through border gates, (10) refusing to grant a definite entry visa to foreigners coming to Vietnam, (11) implementing preventive medicine and carrying out disinfection in public places and locations with suspected and infected cases, (12) zoning and quarantining the areas of suspected and infected cases, and (13) calling and encouraging people to voluntarily quarantine when moving from pandemic areas to Vietnam [27].

In this study, eight key overall policy measures were observed, as described in Box 1 and more detail in Appendix C (see Table A1). First, outbreak announcements and steering documents were published promptly and continuously at all different stages of the pandemic [27,81,82]. Second, there was a series of medical measures implemented, not only benefiting from experience from the SARS epidemic in 2013 [78] but also applying the WHO and the MOH’s guidance. These measures included various types of medical declarations (MDs), from the compulsory level for all people entering the country from 7 March [38,83,84], and even for all people nationwide from 20 March [38,85] to voluntary declarations for all people nationwide from 10 March [86], for all subjects in educational schools and organizations from 26 March [87], and for all domestic travelers from 21 March [88]. Notably, Vietnam was one of the first four countries around the world to culture and successfully isolate the SARS-CoV-2 virus in the early period on 2 February [81,89,90] and produced a detection kit with a capacity of 10,000 sets/day from 4 March [91].

Box 1Policy measures employed in Vietnam to respond to the COVID-19 pandemic.1. Outbreak announcements and steering documents2. Medical measures3. School blockade4. Emergency responses5. Border and entry control6. Social isolation measures7. Financial supports8. Other solutions

Third, school blockade measures included the students of all 63 provinces nationwide having a significantly longer vacation, from early February to 4 May due to the pandemic (three months, whereas the annual summer vacation usually lasts two months). All schools of all 63 provinces nationwide were closed from 6 February [9,92,93,94,95] to the end of April 2020 [96,97,98,99,100]. The MOET also decided to adjust the plan framework twice for the 2019–2020 school year, which delayed the end of the school year and the national high school exam. The second semester program of all general education levels was streamlined to accommodate the extended vacation for the students [82,101]. Fourth, emergency responses such as a series of COVID-19 QRTs established at the ministerial levels for hospitals [9,102,103,104], and many urgent government meetings from the national to provincial levels were implemented immediately, irrespective of day or night, if any NCs were detected that had moved in the community [44,65,105,106,107,108,109,110,111,112].

Fifth and sixth, Vietnam also conducted other strong measures, including border and entry control [9,10,45,78,113,114,115,116,117] and social isolation in many places such as bars, villages, streets, towns, and cities where sources of infection from the patients were confirmed [10,37,39,40,41,43,45,94,118,119,120,121,122,123,124,125,126,127,128,129,130], and even nationwide social isolation [10,131,132].For the first time in history, Vietnam thoroughly implemented all models of isolation: compulsory centralized isolation, home isolation, isolation in place, and even nationwide social isolation. This is considered the most effective measure to limit the spread of the outbreak. Not every country applied the same strict quarantines as in Vietnam (such as nationwide social isolation) [82]. As of 7 April, there were 30 PDs issued by the MOH providing guidance on medical isolation. Tens of thousands of people were sent to concentrated isolation areas (with free support about food, accommodation, medical examination tests, and other living expenses within 14 days, and treatment) [9,82,133]. 

Seventh, in addition, financial supports from the national government (such VNĐ 62 trillion put in place a social security package to help 20 million domestic people affected by the pandemic [134]) and the practical relief activities came from both the governments and the citizens (including the “rice ATM” [135,136,137], free supermarket [138,139,140], etc.) and international communities to make meaningful contributions to the fight against the pandemic in Vietnam. Japan contributed JPY 14 million on 7 February and a test bio-kit worth JPY 4 million on 21 February [141], along with the USA contributing USD 9.5 million [142] and contributions from Korea, France [141]), and others. Finally, a series of other diverse solutions were deployed throughout the country such as strengthening management to avoid collecting and storing goods, advising people about information related to the epidemic through free hotlines and a virtual assistant (chatbot) from the MOH [9], and the appearance of vivid and varied “spiritual dishes” to foster the anti-pandemic universal movement from the boom of music [143,144,145,146], to poetry [147,148,149], films [150,151], etc.

## 4. Discussion and Conclusions

In the current global context, it can be said that no pandemic has had such a profound impact on all aspects of socio-economic life for countries worldwide as COVID-19. Moreover, never in the past have so many unprecedented new policy documents (PDs) been issued widely and continuously in Vietnam to promptly respond to the pandemic [82]. As of 24 July, 2020, Vietnam spent 99 days with no NCs infected from the community [8,9]—365/413 CCs (88.38%) recovered and there were zero deaths [6,7]. To have these impressive numbers, the early preparation and immediate policy responses of government levels from national to provincial, as well as the consensus of all strata, were some of the fundamental contributors to the positive outcomes of these timely and effective measures.

It should be noted that similar to Vietnam, other countries in the region such as Mongolia [152], Taiwan, Hong Kong, Singapore, and South Korea also performed effective policy responses in the early stages of the epidemic [48,49,50]. For example, in Mongolia, the first public preventive measures were introduced by the MOH on 6–12 January, 2020 [152]. Moreover, with the first imported cases on January 13th and January 15th in Thailand and Japan, respectively [153], the Mongolian government immediately held meetings through the MOH’s organization to discuss issues of emergency preparedness regulations and a one-window policy for COVID-19 information from 13–19 January [152]. With rising CCs in China and the appearance of the first CC in South Korea on 20 January, Mongolia issued PDs dated from 20–26 January requiring all educational institutions to be temprarily closed until 30 March and travel restrictions applied to China [152]. In Taiwan, the country established the Enforcement Regulations Governing the Central Epidemics Command Center (Enforcement Regulations—ER, hereafter) in 2004. This is a key component of epidemic preparedness in Taiwan. During each epidemic, with permission from the ER, a Central Epidemic Command Control (CECC) is established by the MOH and welfare, with the approval of the Executive Yuan to “formulate policies, integrate resources, and coordinate responses across different government ministries and agencies” [154]. The CECC is then dissolved once the commanding official confirms that the threat level is low enough, pending approval from the national government. In the context of COVID-19 in 2019–2020, the CECC in Taiwan was established on 20 January—three days before Wuhan city went into lockdown [154]. 

Our analysis demonstrated the multidimensional approaches taken from different relevant government levels (33 agencies from 4 levels from central to local, including, most notably, all 18 ministries and all 63 provinces in the country) in issuing various types of PDs (nine tools: resolution, decision, directive, circular, announcement, plan, dispatch, and guideline) as well as a range of policy measures (eight groups: outbreak announcements and steering documents, medical measures, blockading schools, emergency responses, border and entry control measures, social isolation and nationwide social isolation measures, financial supports, and other measures). As a result of these wide-ranging policy measures and communication tools, issued in a timely manner, the pandemic situation was controlled effectively in the first six months (23 January–24 July). Vietnam’s success in the prevention and fight against the pandemic offers significant lessons for other countries [19,20,23,24,25,26,27,78,79,117,155,156,157].

COVID-19 is at risk of becoming one of the toughest trials for humanity to confront in modern history [158,159]. According to Dr. Tedros Adhanom Ghebreyesus, Director-General of the WHO, COVID-19 is the number one common enemy of humanity, but the actions of many governments and people do not reflect this reality [160]. In the context of much frustration, the policy responses to the COVID-19 pandemic in Vietnam were diverse, proactive, prompt, and widely supported. The proactiveness was not only in legal documents providing guidance before the first two CCs appeared in Vietnam, but also in a series of rapid and drastic larger scale measures that were later deployed synchronously and unprecedentedly [27,81,82]. The most important factor deciding the success of this fight appears to be the great unanimous force of the integration of the whole society, from authorities of all government levels, security forces, and the military to individuals acting together against the outbreak [81,161]. Vietnam is one of the very few countries that mobilized the entire troops, for example, to participate in the pandemic prevention and control from the early days [81]. This mobilization strategy resembles wars that the Communist Party of Vietnam led against the French and American forces in the last century [161]. In particular, the consensus among all classes of people in this war was considered one of the decisive factors in the success [27]. According to one study, Vietnam was the country with the highest satisfaction in the world about the government’s responses to the pandemic (62%) [162,163] and in another, most of those surveyed thought that the solutions of the Vietnamese government were “efficient” and “very effective” [27]. As a result, the country has been praised by governments and scholars across the globe, and within the Vietnamese population.

The strategy of preventing, detecting, tracing, isolating, localizing, and suppressing the pandemic has been thoroughly applied in Vietnam [78,79,80,81,117]. Depending on the specific situation, tactics may change, but the strategy has always been to tighten the defense line for the country against the onslaught of the COVID-19 pandemic [81]. For example, patient CC #17 was hospitalized on 6 March. Two days later, the Deputy PM Vu Duc Dam declared the second phase of the national outbreak [38]. Similar to the first one, marked by the epidemic declaration, the government escalated its public health response to flatten the curve [45], by applying not only compulsory MDs for all people entering the country [38,83], but also voluntary MDs for all people nationwide from 10 March [45,86,164] the same day as the arrival of CC #34, who infected 11 other cases and became the most SCC in period II [38]. Then, the appearance of three F0 CCs (#86, #87, and #91) on 20 March marked the beginning of stage III [45,165,166]. Instantly, the rule of compulsory MDs was put in place for all people nationwide [38,85], and electronic MDs were requested for all domestic travelers from 21 March [88].

In terms of treatment, Vietnam has built and maintained connections from the front line in the concentrated isolation centers to hospitals through QRTs to share, exchange, and support each other in professional work and treatment regimens that are constantly updated for all infectious generations. The application of science and technology in the pandemic prevention and control was accelerated [81], making Vietnam one of four countries that cultured and successfully isolated the SARS-CoV-2 virus in the early period [81,89,90]. The country also conducted research into the production of test kits, proactively produced medical equipment and materials, and promoted the application of information technology in a series of pandemic prevention and control stages (from tracing, following, monitoring, reporting, supporting treatment, etc.) [81]. Vietnam was the first country in the world to apply MDs, and also one of the early adopters of electronic MDs [81]. 

During the pandemic, special policies on education, health care, and social security have been put in place that did not previously exist. Second, besides “COVID-19,” “quarantine” and “isolation” are the most mentioned keywords in Vietnam, with a range of isolation measures put in place. Another highlight is that, despite the economy being seriously affected, the national government has constantly sought to assist all Vietnamese people with the difficulties they are experiencing, so that “no one is left behind” [82,167,168]. This included welcoming every Vietnamese to the homeland, even though on the flights many people carried the virus, which was the main cause in increasing the number of CCs [167], and the unprecedented packages of social security support to reduce difficulties for all classes affected by the pandemic [168]. In addition, in the awareness campaigns for pandemic prevention and control, media forces have never participated together so unanimously and drastically as during this time, from the official newspapers and network operators to social networks [81,168]. With the spirit of publicity and transparency [81,168,169,170,171], information on steering, prevention, and control recommendations as well as outbreak developments has been conveyed quickly to the public to raise public awareness [81,168]. By virtue of these efforts, in the Association of Southeast Asian Nations region, the lockdown measures were lifted first in Vietnam [169] to accomplish a “dual mission”: economic development and COVID-19 pandemic prevention [172]. According to the World Bank, Vietnam’s economy is well placed to recover after the outbreak if it can identify new growth drivers to reinforce the recovery [169].

In summary, the system of COVID-19 policy responses up to 24 July 2020, were effective to defend against the pandemic in Vietnam. PDs were widely and regularly disseminated, which helped to effectively increase public awareness. Promptly prepared and released PDs provided knowledge and measures for all walks of life to prevent and control the pandemic and protect themselves and others in the community. Although not a rich country, Vietnam has become a model for other countries in the fight against the COVID-19 pandemic, with lessons learned in controlling the spread of the outbreak and for overall responses to a public health crisis [26,156,168].

## Figures and Tables

**Figure 1 ijerph-18-00559-f001:**
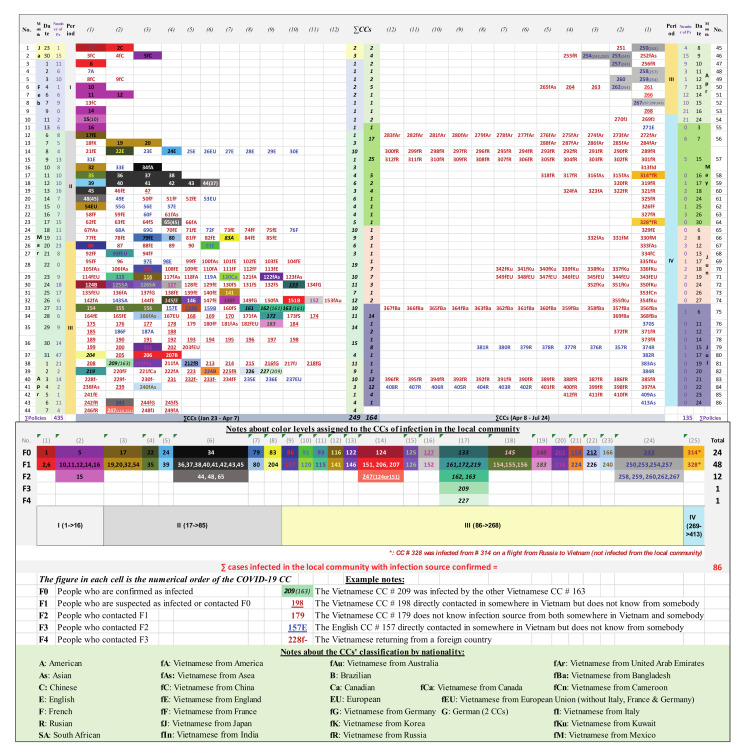
Chronology of COVID-19 confirmed cases (CCs) and the numbers of policy documents (PDs) in Vietnam (as of 24 July 2020).

**Figure 2 ijerph-18-00559-f002:**
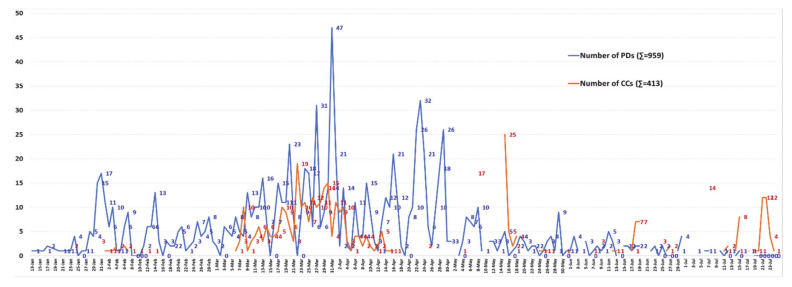
Numbers of CCs and PDs released (as of 24 July 2020).

**Figure 3 ijerph-18-00559-f003:**
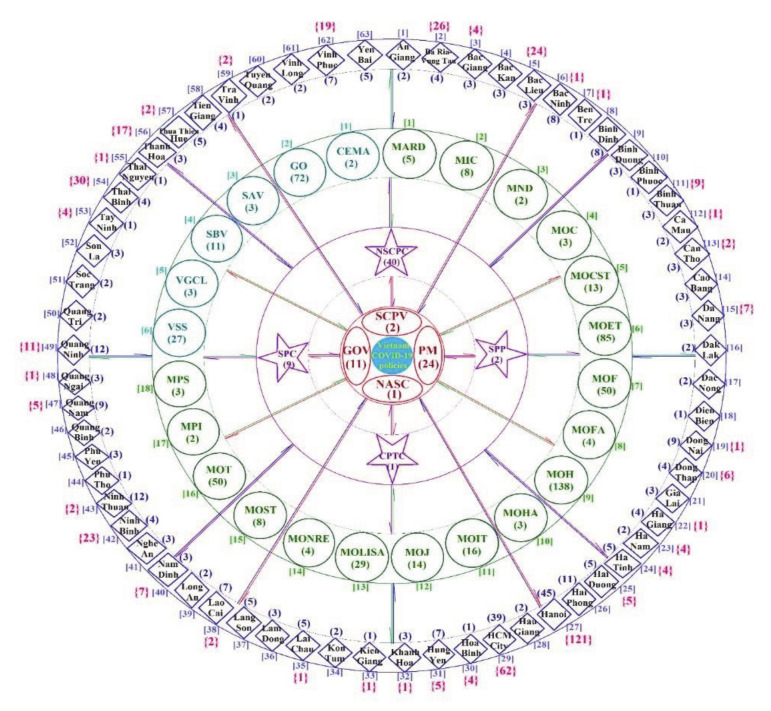
The map of nationwide policy responses to COVID-19 in Vietnam (as of 24 July 2020). (Notes: The number of PDs are in ( ) and the number of CCs of each province are in { }; ordinal number of units due to the alphabet from A–Z at the same level are in [ ]).

**Figure 4 ijerph-18-00559-f004:**
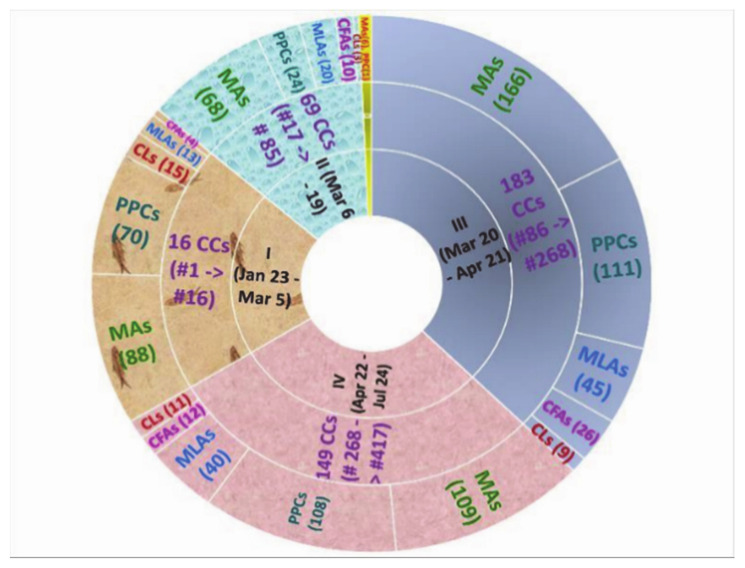
A total of 959 COVID-19 PDs classified by agency groups and timeframe (as of 24 July 2020).

**Figure 5 ijerph-18-00559-f005:**
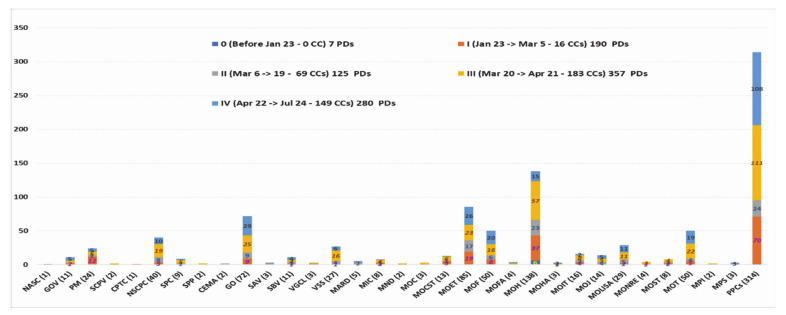
959 COVID-19 PDs classified by government agencies (as of 24 July 2020).

**Figure 6 ijerph-18-00559-f006:**
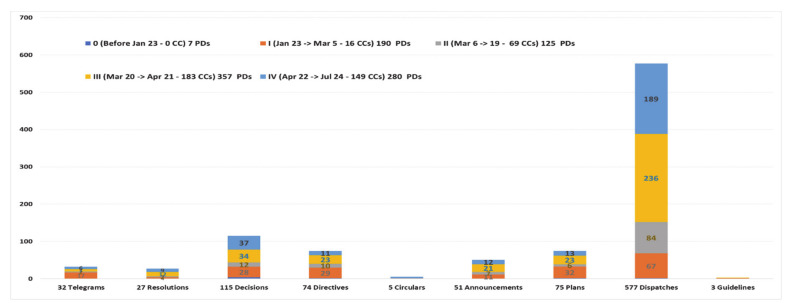
A total of 959 COVID-19 documents classified by policy communication types (as of 24 July 2020).

**Figure 7 ijerph-18-00559-f007:**
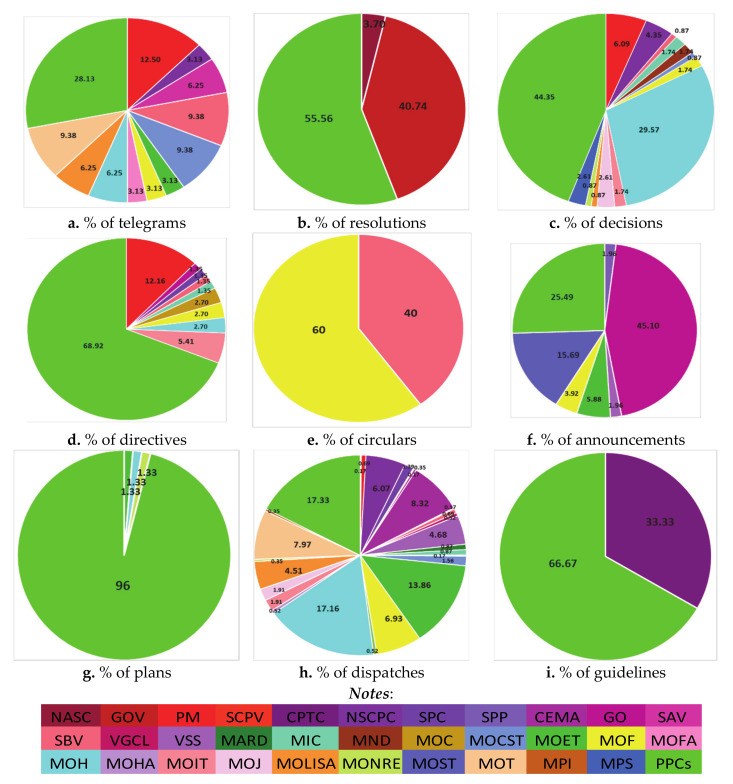
The proportions of COVID-19 policy forms issued by various agencies in Vietnam (as of 24 July 2020).

## Data Availability

Not applicable.

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
