# Peer review of "Policy Responses to the COVID-19 Pandemic in Vietnam"

_ijerph, 2021, doi:10.3390/ijerph18020559_

Round 1
Reviewer 1 Report
Dear authors,
many thanks for the intersting article which shows how the Corona pandemic could be controlled in Vietnam during predominately the first half of the year 2020 supposingly by governmental measures.
However I would like to raise a few points here:
Major:
Shortening the manuscript would be beneficial. It is very hard to get through the manuscript. Some summarizing tables could also help to improve.
Minor:
Figure(s) 7: No abbrevations and the graphics need to be revised (colour and labeling)
Figure 1: Seems to be overwhelmingly full with information. Maybe one may also think of overworking it.
I would also suggest to include at data of the current situation.
Best regards
Author Response
Letter of detailed responses to Reviewer 1
Dear Sir/Madam,
We are grateful for your time and efforts in reviewing our manuscript. Your detailed comments have helped us a lot to improve the quality of our paper.
We have carefully addressed your points in our revised version. Please note that in the revised paper, the parts highlighted in yellow are for correction on the old text and the parts highlighted in green are written a new. Below are our answers to your comments.
Reviewer
(Major) Shortening the manuscript would be beneficial. It is very hard to get through the manuscript. Some summarizing tables could also help to improve.
Response
The revised manuscript was shortened from 15 pages (without appendixes and references) to over 13 pages. For examples, the definitions of 9 PDs communication tools (lines from 382 to 411 in the old version) were moved into a table of Appendix A and some redundant sentences were cut or summarized, such as from line 102 to 108 (in the old version) cut to lines 86 to 88 in the revised version (6 lines to 3) or lines 75 to 86 in the old version were cut in the “Introduction”, etc.
Reviewer
(Minor)
Figure(s) 7: No abbreviations and the graphics need to be revised (colour and labeling)
Response
Figure(s) 7 was edited. The abbrevations were not used in the graphics. Each color was labeled for each agent and this was added in the notes of the end of the figure(s) 7.
Reviewer
(Minor)
Figure 1: Seems to be overwhelmingly full with information. Maybe one may also think of overworking it.
Response
We did not change Figure 1, because it covers the information of the COVID-19 confirmed cases (CCs) from January 23rd to July 24th, 2020 and additional information such date of the CCs, number of policy documents (PDs) issued on the same date of the CCs, the period (I or II or III or IV). This types of information is considered necessary to show the chronology of COVID-19 CCs and the numbers of PDs issued.
Reviewer
(Minor)
I would also suggest to include at data of the current situation.
Response
Data of the current situation was updated as of December 15, 2020 (from lines 60-62 and 68-69 in the revised version).
In closing, we highly appreciate the constructive support and time that you have spent on this manuscript. Your comments have helped improve the quality of our paper considerably. We hope that the revised paper has met your requirements.
Thank you very much
The Study Team

Reviewer 2 Report
The manuscript is fine in all factors used for review consideration. However, the English presentation should be checked for areas of improvement. Examples of this was done for abstract section. The following could be checked for consideration to improve meaning and thought flow:
Line 16: Change "that is" to "has become". If this done "has been" should be removed in Line 17.
Line 36: Add "that could be" before learnt to become " that could be learnt".
Author Response
Letter of detailed responses to Reviewer 2
Dear Sir/Madam,
We are grateful for your time and efforts in reviewing our manuscript. Your detailed comments have helped us a lot to improve the quality of our paper.
We have carefully addressed your points in our revised version. Please note that in the revised paper, the parts highlighted in yellow are for correction on the old text and the parts highlighted in green are written a new. Below are our answers to your comments.
Reviewer
The English presentation should be checked for areas of improvement. Examples of this was done for abstract section. The following could be checked for consideration to improve meaning and thought flow:
Line 16: Change "that is" to "has become". If this done "has been" should be removed in Line 17.
Line 36: Add "that could be" before learnt to become " that could be learnt".
Response
The whole manuscript was checked carefully to improve the quality of English presentation. A lot of points from word to phrase and paragraph level were edited, such as “has become” for “that is” (line 15), then “has been” deleted (line 16); “that could be learnt” added to replace for “which could offer lessons” (line 34), the definitions of 9 PDs communication tools (lines from 382 to 411 in the old version) were moved into a table of Appendix A and some redundant sentences were cut or summarized, such as from line 102 to 108 (in the old version) cut to lines 86 to 88 in the revised version (6 lines to 3) or lines 75 to 86 in the old version were cut in the “Introduction”, lines 77 – 81 (in the old version) cut, etc.
In closing, we highly appreciate constructive support and time that you have spent on this manuscript. Your comments have helped improve the quality of our paper considerably. We hope that the revised paper has met your requirements.
Thank you very much
The Study Team

Reviewer 3 Report
This paper presents an interesting policy response to covid-19. The article provides a case study about the policy responses to the COVID-19 pandemic in Vietnam. The paper provides a piece of useful information about covid-19 response to the pandemic in Asia. This is a well-organized paper and the data review and results are new and useful for the scientific community. This paper has enough attributes to be considered as a good candidate for publication after revision. While this paper offers some interesting data and results, this paper needs to be revised according to the following suggestions:
Although the introduction is a good description of Covid-19 development and infection, it is too long and redundant. For example, lines 75 to 86 and lines 102 to 108.
Figures should match the excel page in the supplementary file or the figures should be break into a clear graphic (the graphic is too small for reading and hard to find in the supplementary files).
Lines 266 to 285, how are these paragraphs related to the urgent and timely policy responses implemented by all the relevant authorities in Vietnam? These lines should be moved to the conclusions.
The government of Vietnam implemented perfect and timely policies to control the Covid-19 infection and spread, but How do people respond to these implemented policies? Do These implement policies were compulsory? How does the economy was impacted by these implemented polices to control the Covid-19 spread? What was the socio-economic plan or policy response and how these policies were implemented in Vietnam?
Author Response
Letter of detailed responses to Reviewer 3
Dear Sir/Madam,
We are grateful for your time and efforts in reviewing our manuscript. Your detailed comments have helped us a lot to improve the quality of our paper.
We have carefully addressed your points in our revised version. Please note that in the revised paper, the parts highlighted in yellow are for correction on the old text and the parts highlighted in green are written a new. Below are our answers to your comments.
|
Reviewer |
Response |
|
“Although the introduction is a good description of Covid-19 development and infection, it is too long and redundant. For example, lines 75 to 86 and lines 102 to 108” |
In general, the whole manuscript was checked carefully to make the content more concise. A lot of points from word to phrase and paragraph level were edited, such as “has become” for “that is” (line 15), then “has been” deleted (line 16); “that could be learnt” added to replace for “which could offer lessons” (line 34), the definitions of 9 PDs communication tools (lines from 382 to 411 in the old version) were moved into a table of Appendix A and some redundant sentences were cut or summarized, such as from line 102 to 108 (in the old version) cut to lines 86 to 88 in the revised version (6 lines to 3) or lines 75 to 86 in the old version were cut in the “Introduction”, lines 77 – 81 (in the old version) cut, etc. Therefore, the revised manuscript was shortened from 15 pages (without appendixes and references) to over 13 pages. |
|
Q2. “Figures should match the excel page in the supplementary file or the figures should be break into a clear graphic (the graphic is too small for reading and hard to find in the supplementary files).” |
A2. The excel sheets were renamed and ordered according to the figures in the paper to have more clear information for reading.
|
|
Q3. Lines 266 to 285, how are these paragraphs related to the urgent and timely policy responses implemented by all the relevant authorities in Vietnam? These lines should be moved to the conclusions. |
We made the change according to your suggestion, the lines 266-285 in the old version were moved to the lines 419-437 (in the revised version) of the conclusions |
|
Q4. “The government of Vietnam implemented perfect and timely policies to control the Covid-19 infection and spread, but How do people respond to these implemented policies? Do These implement policies were compulsory? How does the economy was impacted by these implemented polices to control the Covid-19 spread? What was the socio-economic plan or policy response and how these policies were implemented in Vietnam?” |
a4. - “How do people respond to these implemented policies?” We provided some additional information according to your suggestion. In general, most people living in Vietnam responded to the COVID-19 policies very positively. This was added at some points of the paper. Many movements were taken place actively in the communities, schools, or organizations to fight the pandemic, such as “the appearance of vivid and varied “spiritual dishes” to foster the anti-pandemic universal movement from the boom of music [143–146], to poetry [147–149], films [150,151], etc.” (lines 407-409 in the revised version), or “the practical relief activities, that come from both the governments and the citizens (including “rice ATM” [135–137], free supermarket [138–140], etc.)” (lines 401-402 in the revised version), “The most important factor deciding the success of this fight appears to be the great unanimous force of the integration of the whole society, from authorities of all government levels, security forces, the military to individuals to act together against the outbreak [81,161].” (lines 455-457 in the revised version) or “In particular, the consensus among all classes of people in this war was considered one of the decisive factors to the success [27]” (lines 461-462 in the revised version), etc. - “Do These implement policies were compulsory?” The COVID-19 policy in general is compulsory in many policy documents (PDs) with the important issues, for example “compulsory centralized isolation” (line 392 in the revised version), "nationwide social isolation" (lines 393, 395 and 444 in the revised version), “blockade the schools” (lines from 353-354, 371-372, 442 in the revised version), medical declarations (MDs) (lines 369, 473, 477), “border and entry control” (lines 388, 443 in the revised version), etc. - “How does the economy was impacted by these implemented polices to control the Covid-19 spread?” This is not a basic content of the study. However, in Vietnam, at least “20 million domestic people affected by the pandemic [134])“ (line 400 in the revised version), “financial supports from the national government (such VNĐ 62,000 billion social security package” (lines 399-400 in the revised version) through some legal documents conducted that is a typical example to point out that the livelihood of many vulnerable people (affected by the pandemic), was supported by some of the COVID-19 policy measures implemented. This was, once again, affirmed in a larger context, that the national economy is predicted to restore, thanks to the implementation of effective pandemic prevention and control measures. “According to the World Bank, Vietnam’s economy is well placed to recover after the outbreak if it can identify new growth drivers to reinforce the recovery [169].” (lines 504-505 in the revised version). - “What was the socio-economic plan or policy response and how these policies were implemented in Vietnam?” This is also not a reviewed content of the study because the socio-economic plans and policy responses were mainly implemented after Ausgust, 2020. However, the stage IV (Apr 22-Jul 24) in Vietnam also showed that some socio-economic measures were begun to be implemented to ensure economic restoration (lines 179-180), “to accomplish a "dual mission", both economic development and COVID-19 pandemic prevention [172].” (lines 503-505 in the revised version) |
In closing, we highly appreciate constructive support and time that you have spent on this manuscript. Your comments have helped improve the quality of our paper considerably. We hope that the revised paper has met your requirements.
Thank you very much
The Study Team
